# Quartz Crystal Microbalance with Dissipation Monitoring of Dynamic Viscoelastic Changes of Tobacco BY-2 Cells under Different Osmotic Conditions

**DOI:** 10.3390/bios11050136

**Published:** 2021-04-27

**Authors:** Zongxing Chen, Tiean Zhou, Jiajin Hu, Haifeng Duan

**Affiliations:** 1College of Bioscience and Biotechnology, Hunan Agricultural University, Changsha 410128, China; zxc_0914@163.com (Z.C.); jjhu66@hunau.edu.cn (J.H.); dhf77882021@163.com (H.D.); 2Hunan Provincial Engineering Technology Research Center for Cell Mechanics and Function Analysis, Changsha 410128, China

**Keywords:** QCM-D, BY-2 cell, viscoelasticity, osmotic stress, morphology

## Abstract

The plant cell mechanics, including turgor pressure and wall mechanical properties, not only determine the growth of plant cells, but also reflect the functional and structural changes of plant cells under biotic and abiotic stresses. However, there are currently no appropriate techniques allowing to monitor the complex mechanical properties of living plant cells non-invasively and continuously. In this work, quartz crystal microbalance with dissipation (QCM-D) monitoring technique with overtones (3–9) was used for the dynamic monitoring of adhesions of living tobacco BY-2 cells onto positively charged N,N-dimethyl-N-propenyl-2-propen-1-aminiumchloride homopolymer (PDADMAC)/SiO_2_ QCM crystals under different concentrations of mannitol (C_M_) and the subsequent effects of osmotic stresses. The cell viscoelastic index (CVI_n_) (CVI_n_ = ΔD⋅n/ΔF) was used to characterize the viscoelastic properties of BY-2 cells under different osmotic conditions. Our results indicated that lower overtones of QCM could detect both the cell wall and cytoskeleton structures allowing the detection of plasmolysis phenomena; whereas higher overtones could only detect the cell wall’s mechanical properties. The QCM results were further discussed with the morphological changes of the BY-2 cells by an optical microscopy. The dynamic changes of cell’s generated forces or cellular structures of plant cells caused by external stimuli (or stresses) can be traced by non-destructive and dynamic monitoring of cells’ viscoelasticity, which provides a new way for the characterization and study of plant cells. QCM-D could map viscoelastic properties of different cellular structures in living cells and could be used as a new tool to test the mechanical properties of plant cells.

## 1. Introduction

Changes in the extracellular microenvironment will cause changes in plant cell structure and physiological function, which will inevitably lead to changes in the mechanical properties within and between cells. These mechanical interactions have very important physiological and pathological significance. The study of animal cell mechanics has entered the mainstream of science [1,2,3], however, most implementations (e.g., atomic force microscopy and cellular force microscope) are only suitable for single cell measurements and require some type of forces to be applied to the measured cells, which preclude their continuous and long-term measurements of massive living plant cells of statistical significance.

In comparison with animal cells, the cell wall and vacuoles of plant cells are the main factors that hinder real-time monitoring. For animal cells, there is a continuum of the extracellular matrix (ECM)-integrin-cytoskeleton, which is demonstrated by experiments to play an important role in the transmission and transformation of force signals [4]. It is considered that the cell wall of plants is a dynamic and sensitive structure, which forms a continuum with the plasma membrane and cytoskeleton, and can feel the changes in the external environment and respond accordingly [5]. It is realized that mechanical measurements are most informative regarding cell (wall) structures [6].

Cellular structures, including cell walls and cytoskeletons are constantly changing and/or remodeling under biotic and abiotic stresses [7,8]. The degrees of damage to cell walls, the wall-plasma membrane-cytoskeleton continuum, the cytoskeleton and their recoveries would indicate plant cells’ health status and their resistance to these stresses. For example, wall remodeling under abiotic stresses involves the formation of reactive oxygen species (ROS) and peroxidases, which cross-link phenolic compounds and glycoproteins of the cell walls causing stiffening. If ROS-levels remain high during prolonged stress periods, OH^·^-radicals are formed which lead to polymer cleavage causing softening [9]. As minute changes in wall structure leading to the changes in wall’s stiffness or viscoelasticity cannot be observed by optical microscopy, it is not surprising that little is known about changes in the cell wall itself, a challenge for a better understanding of abiotic stress tolerance. It is even more difficult to observe the structural changes of plant cytoskeleton, fluorescent labelling of actin and/or microtube has to be made, however, it may suffer from photobleaching and fluorescent dye losses over long measurement periods, leading to ambiguous results [10].

As a label-free, non-invasive, and highly sensitive technique, the quartz crystal microbalance (QCM) has been widely used in many fields as reviewed in several publications [11,12,13,14,15,16]; including biomolecular detection [11], cell research [11,12], drug product development [13], phase transitions of biomimetic membranes [14], bacterial adhesion [15] and infectious diseases [16]. Here we only present some of the major progresses of the QCM technique in cellular and biomolecular assays in the last 4–5 years, summarized in Table 1.

In this study, a strong cationic polyelectrolyte, N,N-dimethyl-N-propenyl- 2-propen-1-aminium chloride homopolymer (PDADMAC) was used to adhere the negatively charged cell wall of tobacco BY-2 cells to the QCM chip. The dynamic viscoelastic changes of living BY-2 cells were characterized by cell viscoelastic index (CVI) [24], we used the technique of multiple overtones and wide frequency (F) quartz crystal microbalance with dissipating (D) monitoring (QCM-D) to distinguish different depths of cellular structures based on their viscoelastic differences and to monitor the viscoelastic changes of BY-2 cells under different osmotic stresses.

## 2. Materials and Methods

### 2.1. Materials

Mannitol is a kind of plant tissue compatible solute, which is commonly used in the study of plasmolysis and as a regulator of osmotic pressure for plant cells, and has no other effect on plant cells. PDADMAC is a strong cationic polyelectrolyte, it was previously used to modify QCM chips to interact with negatively charged protoplasts [25].

### 2.2. Sample Preparation

BY-2 cells were kindly provided by Prof. Xuewen Zhang of College of Bioscience and Biotechnology at Hunan Agricultural University. The cells were cultured in MS medium and containing sucrose (3%), KH_2_PO_4_ (0.2 g/L), VB_1_ (1 mg/L), 2-4-D (0.2 mg/L), inositol (0.1 g/L), glycine (2 mg/L), pH = 5.8. They were grown at 26 °C under dark culture and constant agitation (105 rpm). The BY-2 cells were sampled for measurements on the fourth day. Before being sampled, a 100 objective cells sieve was used to filter BY-2 cells. BY-2 cells were suspended in certain concentration (0.2 M–0.4 M) of mannitol solution for 2 h at 26 °C, in order to keep the cells into the required states.

### 2.3. Quartz Crystal Microbalance with Dissipation Monitoring

QCM-D measurements were performed with a Q-Sense E4 unit (Q-Sense AB, Göteborg, Sweden) by simultaneously monitoring shifts in frequency (ΔF) and energy dissipation (ΔD) of a 5 MHz silica-coated QCM-D crystal (QSX-303) at different overtones. AT-cut crystals (Q-Sense) with 14 mm diameters and 50 nm thermally evaporated silicon oxide coats were used for all QCM-D experiments. Before the test, the silica-coated crystal surface was treated by plasma cleaner (PDC-32G Compact Desktop Plasma Cleaner, Harrick Plasma, Ithaca, NY, USA) with argon gas at high lever for 30 s, then soaked in 1% SDS for 30 min, cleaned with DI water (Milli-Q Advantage A10, Millipore, Shanghai, China) and dried with compressed nitrogen, then treated again by plasma cleaner with argon gas for 30 s. The cleaned crystal was assembled into the reactor followed by modification of the crystal with 0.1 mg/mL PDADMAC for 30 min, then cleaned with DI water and dried with compressed nitrogen. The exponential decay of the crystal oscillation was recorded and analyzed, yielding the frequency and dissipation changes at frequencies of 15, 25, 35, 45, and 55 MHz. The data presented in the main text was recorded at the third overtone (15 MHz), fifth overtone (25 MHz), seventh overtone (35 MHz), and the ninth overtone (45 MHz), and then normalized based on the overtone number.

A 200 mL suspension of 4 × 10^4^ BY-2 cells was incubated on each PDADMAC (0.1 mg/mL)-modified quartz crystal for 2 h (26 °C), prior to being mounted into a clean open module (QFM401, Q-Sense). After the BY-2 cells immobilized QCM crystal and (PDADMAC)/SiO_2_ modified reference QCM crystal were incubated in the same concentration of mannitol solution, the mannitol solution was replaced with mannitol solution of different concentration sequentially. ΔD·n and ΔF were monitored following the changes under different osmotic conditions (Figure 1a–d).

### 2.4. Optical Microscope Observation

Simulations of QCM pretreatments were carried out in glass culture plates (diameter 14 mm, thickness 0.085–0.115 mm, In Vitro Scientific, Hangzhou, China). The plates were treated by plasma cleaner with gas argon at high step for 1 min, the glass bottoms of the plates were modified with 0.1 mg/mL PDADMAC for 30 min, then cleaned with DI water and dried with compressed nitrogen. In the first group, BY-2 cells were suspended in a certain concentration of mannitol solution (C_M_) at 26 °C for about 2 h, and then the cells were placed under the microscope (IX 71, Olympus, Shenzhen, China) to observe the cell morphology. In the second group, BY-2 cells were immobilized on PDADMAC modified glass plate and then stayed in the mannitol solution for 2 h at 26 °C. The concentration of the mannitol solution was changed every 2 h, then the morphology of the immobilized BY-2 cells was observed by an optical microscope (Olympus IX 71). In the third group, we used a Lumascope 720 microscope (Etaluma, Carlsbad, CA, USA) that automatically took pictures to record changes in cell morphology and set the microscope to automatically take pictures every 5 s for 35 min. BY-2 cells were suspended in the isotonic solution for about 1 h, after automatic photographing, 1 M mannitol solution was used to change the mannitol concentration to 0.4 M.

## 3. Results

### 3.1. Real-Time QCM-D Responses during Adhesions of BY-2 Cells

Based on the conclusions of Nam-Joon [26], if the induced QCM changes in frequency (ΔF) and dissipation (ΔD) at different overtone (n) by the coupled substance obey the following rule, ΔD_r_·n^0.5^ and -ΔF_r_/n^0.5^ are constants for different overtones, then this substance is a pure Newton fluid, if not, this substance has viscoelastic feature.

There were changes in both frequency and dissipation for each overtone following adhesion of BY-2 cells and change of mannitol solution (Figure 1a,b). Once the BY-2 cells were added, the QCM’s frequency F dropped and dissipation D increased promptly and then gradually stabilized which lasted for at least 1 h. The change of mannitol concentration itself also induced observable changes of D and F for the reference crystal without cells, but F and D became stable in a very short period of time (Figure 1c,d).

In different concentrations of mannitol, the beginning stage of the adhesion process of BY-2 cells show obvious changes in both frequency and dissipation for each overtone (Figure 2). The most obvious changes were in overtones 3 and 5, the length of the adsorption was about 1 h. In order to accurately show the changes of frequency and dissipation induced by the adhesions of BY-2 cells and osmotic stresses, we used ΔD_r_ and ΔF_r_ to represent the changes in cells relative to that of the reference crystal in mannitol, and ΔD_r_ = ΔD_BY-2_ − ΔD_Mannitol_, ΔF_r_ = ΔF_BY-2_ − ΔF_Mannitol_. As shown in Figure 1g,h, at the same osmotic pressure, ΔD_r_·n^0.5^ and ΔF_r_/n^0.5^ were not constants for different overtones indicating the viscoelastic feature of living BY-2 cells, and which were different from the Newtonian fluids.

### 3.2. Dynamic Viscoelastic Changes of BY-2 Cells under Osmotic Stresses

According to the characteristics of the Q-Sense signal, different overtones detect different depths of the coupled materials. The CVI of each overtone had its own changing characteristics. For the third overtone (n = 3), the change of CVI was most obviously due to the higher detection depth to the cell body. There were some special mannitol concentrations at which CVI were extremes. Figure 3 showed that these extremes were around the initial plasmolysis concentration of mannitol (C_ip_) when the overtones n were 3 and 5. When C_M_ was close to C_iP_ (0.3 M, Figure 3a), the QCM had the highest sensitivity. When the initial mannitol concentration was 0.4 M, the deplasmolysis phenomenon was conspicuous (Figure 3a). And, when the initial mannitol concentration was 0.3 M, the plasmolysis phenomenon was conspicuous (Figure 3c and Figure 4). When n was 7 and 9, the main trend of CVI was that it decreased as the increase of C_M_, because in these higher overtones (n = 7, 9), the Q-Sense may only detect the region of the cell wall, and while C_M_ increased, the force of protoplast exerted on cell wall decreased.

When 0 M < C_M_ < C_iP_, as the decrease of C_M_, CVI increased. However, in pure water (C_M_ = 0 M), CVI decreased instead, implying that the wall elasticity was no longer maintained and the cell’s viscoelasticity was governed by the cytoskeleton, while when 0.7 M > C_M_ > C_iP_, as the increase of C_M_, the CVI-C_M_ relationship became more complicated. For higher overtones (n = 7, 9), CVI usually monotonically decreased as the increase of C_M_. When C_M_ > 0.7 M, the protoplasts lost so much water that the internal structures, such as the cell skeleton of the cells might be irreversibly changed, due to the loss of water and energy.

The mechanical properties of plant cells are not homogeneous, that is, different structures, including cell walls, cell membranes and cytoskeletons have different mechanical properties. In particular, the cell wall and cytoskeleton mainly determine the mechanical strength and morphology of the cell. By changing the frequency of the QCM crystal, the sound wave penetration depth can be modulated, this can be achieved by Q-sense instrument which provides wide frequency measurement through different overtones. The smaller the overtone, the lower the frequency, the deeper the penetration depth of sound wave, allowing the study of the cytoskeleton and even the nucleus. The larger the overtone n, the higher the frequency, the thinner the penetration depth, and only reaching the cell/sensor interface layer up to the cell wall or protoplast cell membrane. The average viscoelasticity corresponding to the penetration depth at overtone n is semi-quantitatively characterized by cell viscoelastic index CVI_n_. Figure 3a,c show the dynamic viscoelastic changes of BY-2 cells of normal state around isotonic pressure being subjected to hypotonic and hypertonic stresses respectively. At constant n which corresponds to specific cell structure(s), CVI_n_ changed as the mannitol concentration or osmotic pressure changed. At hypotonic pressures, CVI_7_ > CVI_9_ > CVI_5_ > CVI_3_ or |CVI_7_| < |CVI_9_| < |CVI_5_| < |CVI_3_|, CVI_7_ was the highest over the whole concentration range of C_M_. In comparison, CVI_9_ which represents the viscoelasticity of cell wall was always only smaller than that of CVI_7_. We speculate that CVI_7_ detects both cell wall and protoplastic membrane, under hypotonic pressure, the cell membrane would be subjected to tensile stress (turgor pressure) which makes the membrane stiffer and we conclude that at hypotonic pressures, the stiffness of different cellular structures is in the following order: protoplast membrane > cell wall > cytoskeleton.

However, when the BY-2 cells were maintained at higher osmotic pressure of 0.6 M mannitol for 2 h before the mannitol concentration was gradually decreased, we found different viscoelastic changes corresponding to different cellular structures (Figure 3b) compared to those BY-2 cells initially maintained at 0.4 M mannitol. When the BY-2 cells were incubated at higher osmolality of 0.6 M mannitol for 2 h, the cytoplasmic wall separation was more serious. Some structures between the cell membrane and the cell wall were more severely damaged. Figure 3b showed that CVI_7_ was no longer the largest at 0.45 M, and the largest was CVI_3_. The results indicated that the recovery of the cytoskeletal structure of the BY-2 cells was much faster than those of the cell membrane and the cell wall, making the cytoskeleton the hardest structure.

In order to see and analyze more clearly the transient responses of BY-2 cells during the initial periods of hypertonic stresses, the earlier 0.5 h or so responses of Figure 3c were expanded and re-presented as Figure 4. Once the mannitol concentration was increased from 0.3 M to 0.4 M, the cells had an obvious and transient softening, then became hardening followed by a second softening (Figure 4a). At 0.3 M mannitol, the cells were in a hypotonic state with a certain turgor pressure, therefore, all the CVIs > 0, once C_M_ was increased to 0.4 M of hypertonic state, the cells lost their pressure, CVIs changed from positive to negative, resulting in the first softening. Then as the progress of plasmolysis, the tensile stress exerted on the plasma membrane through the Hechtian strands due the contractions of the protoplasts [27] made all the cellular structures harder as evidenced by the increased CVIs, indicating that most of the Hechtian strands were intact and the cell wall-plasma membrane-cytoskeleton continuums were relatively complete. This hardening process lasted about 1-2 min followed by a second softening. The cytoskeleton structure depolymerized and re-polymerized during the plasmolysis as demonstrated by the initial drops in CVI_3_ and CVI_5_ followed by their increases, whereas the cell wall and plasma membrane became stable after their softening during the plasmolysis as indicated by the initial drops in CVI_9_ and CVI_7_ followed by their stable responses.

When C_M_ was increased to 0.5 M and 0.6 M, the plasmolysis was enhanced and the cells were in the dehydration status (Figure 5a7,b7,a8,b8). The CVIs (Figure 4b,c) had similar responses as Figure 4a except for that the first softening accompanied with the loss of turgor pressure was absent. Most of the protoplasts were still maintained in tight connections with the cell walls at 0.5 M mannitol (Figure 5a7,b7), however, when C_M_ was increased to 0.6 M, severe shrinkages of the protoplasts were observed (Figure 5a8,b8), leading to the further loss of the Hechtian strands and the cell wall became the softest cellular structure as indicated by the lowest CVI_9_.

As the further increase of mannitol concentration C_M_ to 0.7 M and 0.8 M (Figure 4d,e), dehydration of the protoplasts became more serious and the Hechtian strands gradually disintegrated. Again, all CVIs increased first then decreased at these two mannitol concentrations, significant (0.7 M) and minor (0.8 M) depolymerizations of the cytoskeletons were observed followed by their recoveries. Under the actions of membrane tension and cytoskeleton reorganization, the cells gradually stabilized, and the protoplasts tended to be spherical.

When C_M_ was increased to 0.9 M, all the cellular structures became slightly harder under the plasma membrane tension then became softer (Figure 4f); however no further increase of CVI_3_ was observed indicating that the cytoskeletons lost their function of re-polymerization. The volumes of the protoplasts decreased only a little bit (Figure 5a11,b11).

When C_M_ reached the highest concentration of 1.0 M tested, the volumes of the protoplasts had little change (Figure 5a12,b12). Almost all the Hechtian strands were completely lost, there was no more hardening of cellular structures caused by plasma membrane tension. Indeed, all the CVIs were much smaller than those of lower concentrations of mannitol. The cellular structures were irreversibly damaged.

### 3.3. Multi-Structural Changes of BY-2 Cells under Different Mannitol Concentrations

Microscope observation results showed that the mannitol concentration that induced initial plasmolysis (C_ip_) for BY-2 cells is between 0.3 M and 0.35 M. In pure water (0 M), BY-2 cells swelled up and busted due to absorption of too much water and the protoplast membrane could not maintain its elasticity without breaking (Figure 5a1,b1). When mannitol concentration (C_M_) was lower than C_ip_ (Figure 5a1–a4,b1–b4), BY-2 cells became swollen and expanded. When C_M_ > C_ip_ (Figure 5a5–a11,b5–b11), BY-2 cells dehydrated and under different degrees of plasmolysis. In 1 M mannitol concentration, most cells died due to serious dehydration (Figure 5a12,b12).

In order to observe the process of plasmolysis more intuitively, we recorded the process of plasmolysis with a Lumascope 720 microscope which can automatically photograph it. At the moment of adding hyperosmotic solution, the change of osmotic pressure made the cells lose water, the vacuole became smaller, and plasmolysis occurred immediately. With the passage of time, the process of plasmolysis increased and reached a stable state in about 30 min (Figure 6). The protoplasts were not completely detached from the cell wall, and there were some cell membranes connected to the cell wall, and some parts were only punctate connected, and the protoplasts were irregularly shaped, the plasma membrane separated from the cell wall with the formation of several concave pockets.

In isotonic solution, the cell membrane was tightly connected with the cell wall without voids. When 1 M mannitol solution was added, the osmotic pressure of the solution increased, the cells lost water and the cell membrane became irregularly crumpled and the cell wall was deformed by the action of Hechtian strands between the cell membrane and the cell wall while this depression of the cell wall recovered in about 30 min (the position indicated by the red arrow). As time went on, the degree of wall separation increased and finally became stable (the position indicated by the blue arrow).

## 4. Discussion

The QCM technique can be applied to study the mechanical properties of plant cells, as long as we can modify the QCM chip with an appropriate substance allowing the plant cells to adhere. Our experimental results showed that living BY-2 cells have viscoelastic feature and their viscoelasticity changed under different osmotic conditions. Q-sense is a convenient instrument for measuring the characteristics of the live plant cells, and by measuring the changes at different overtone frequencies showing the information of different depths of plant cell and allowing more comprehensive and real-time monitoring of subtle changes of the cell.

### 4.1. The Cell Viscoelastic Index CVI

According to the formula [26], the cell viscoelastic index CVI (CVI = ΔR/ΔF) [24] is modified as CVI = ΔD·n/ΔF to reflect the effect of the overtone number and the relationship between motional resistance (R) and dissipation (D). Normally during the adhesion of cells, QCM frequency F decreases, R or D increases, therefore CVI is negative, the smaller the CVI, the softer the cells; however, there are some situations where F would increase instead due to the existence of stress exerted on the QCM surface, then CVI is positive, and the smaller the CVI, the harder the cells. In any case, the smaller the ǀCVIǀ, the harder the cells. The relative ΔD and ΔF changes in BY-2 cells adhered crystal versus the reference crystal was used to verify the viscoelastic feature of BY-2 cells (Figure 2) and to calculate CVI to follow viscoelastic changes under different osmotic conditions (Figure 3).

As the BY-2 cells have irregular shapes, there are interactions between the cells, and it takes a certain amount of time to establish stable interaction between cells and PDADMAC, so the adhesion of BY-2 cells onto PDADMAC took about 1 h. The adsorption of BY-2 cells led to the decrease of the frequency and the increase of the dissipation. The results also showed that the BY-2 cells are of viscoelastic feature, which is different from that of the pure Newtonian fluid (Figure 1 and Figure 2). The state of cells changes with the mannitol concentration which can be traced by the QCM’s responses. As both the cells’ absorption and lose of water are processes, so the cells slowly changed their shapes resulting in changes in QCM’s responses. Either of these processes needs enough time to reach a steady state, so after the concentration of mannitol was changed, the changes in QCM’s responses of the cell’s group took much longer time (about 1.5 h) to reach a relatively stable state, whereas the control group only took about 20 min.

According to the characteristics of the Q-Sense signal, the lower the overtone, the deeper depth of the device can detect, which allows QCM to detect both the cell wall and the protoplast, and relatively easy to detect the plasmolysis phenomenon.

### 4.2. Characteristics of Cell Viscoelasticity under Hypertonic Conditions

Due to the aggravation of plasmolysis caused by water loss, protoplast would produce a certain tensile stress on the plasma membrane by the action of Hechtian strands [27], which is expected to increase the hardness of the stressed cellular structures according to the model of cellular tensegrity [28], this model is widely accepted for describing animal cells, and was also proposed to be appliable to plant cells [29]. Our work described here verified experimentally the following two important predictions in plant cell mechanics. (1) Under the status of dehydration (hypertonic stress), plasma membrane tension may be increased as in hypotonic condition. (2) The increased plasma membrane tension would increase the hardness of the cellular structures assuming cellular tensegrity is also applicable to plant cells. As shown in Figure 3c and more clearly in Figure 4, transient increases in the hardness of the cellular structures of BY-2 cells occurred when they were subjected to the hypertonic stresses of 0.4–0.9 M concentrations of mannitol.

Moreover, the wide frequency QCM-D technique with multiple overtones allowed us to analyze and compare the dynamic viscoelastic changes of different cellular structures. First of all, the plasma membrane was always the hardest cellular structure being directly subjected to the actin cytoskeleton-generated tensile stress, which was demonstrated by the fact that CVI_7_ was always the largest among all the CVIs. Then CVI_9_ which characterizes the cell wall should be the next hardest cellular structure, however, this is only valid when the cell wall-plasma membrane-cytoskeleton continuum is maintained. CVI_9_ was larger than CVI_3_ and CVI_5_ only at lower mannitol concentration of 0.4 M, then CVI_7_ became close to, even smaller than CVI_3_ and CVI_5_ as the increase of C_M_. This can be ascribed to that as the increase of C_M_, the more loss the Hechtian strands and the less complete the cell wall-plasma membrane-cytoskeleton continuum, resulting in less force transferred to the cell wall. The unique of the cytoskeleton structure lies in its capability of disassembly and assembly through the de-polymerization and polymerization of cytoskeleton components of microtube and actin filaments. We observed the phenomena of cytoskeleton disassembly and assembly when C_M_ was increased from 0.4 M to 0.8 M indicated by the dynamic changes of CVI_3_, thereafter, the cytoskeleton lost its function of disassembly and assembly as C_M_ was further increased. When C_M_ was increased to 1 M, the transient increase in the hardness of the cellular structures disappeared due to the complete loss of the Hechtian strands and the disfunction of the cytoskeleton, and all the CVIs were much smaller than those at lower concentrations of C_M_.

A plant’s response to drought stress depends on the specific strain caused by stress in the plant. Zhang and Bartels distinguished dehydration tolerance from desiccation tolerance, an essential distinction for phenotypic and interpreting survival and recovery [30]. Survival due to dehydration tolerance, is expressed by delayed mortality driven mainly by a more resilient plant metabolism, a feature typical of resurrection plants [31]. Figure 3d showed the dehydration tolerance process of BY-2 cells in dehydration followed by recovery to normal state. In ‘hydrated’, ‘mild stress’ and ‘moderate stress’ states, cells were generally able to return to the functional state of water absorption and vitality. In ‘severe stress’ and ‘desiccation’ states the cells were severely dehydrated and unable to restore vitality (Figure 5a10–a12,b10–b12).

An intriguing phenomenon is that the initial mannitol concentration the BY-2 cells is subjected to determines the sensitivity of plasmolysis detection. When the initial C_M_ was close to C_ip_, the cell function was in normal state, and the protoplast could be separated from the cell wall gently, and the Q-Sense could detect the obvious signal of the separation.

### 4.3. Characteristics of Cell Viscoelasticity under Hypotonic Conditions

When 0 M < C_M_ < C_iP_, as the C_M_ decreases, the extracellular osmotic pressure was lower than the cells, so protoplasts exert more force on the cell wall and made the cell wall stiffer and CVI increased. In this case, the increase in CVI indicated that the cell wall mainly determined the viscoelastic properties of BY-2 cells. However, at the special point C_M_ = 0 M mannitol, most of the protoplasts busted, the turgor disappeared, the rigidity of the cell wall descended, while flexibility increased. At the same time, the contents of the protoplasts would leak out into the buffer solution, so the viscoelasticity of the buffer solution was changed. Some special factors contributed to the decrease in CVI and the results were not only induced by the viscoelasticity of cell walls. Another intriguing phenomenon was that when the initial mannitol concentration C_M_ < C_ip_, the cell was in the swelling state (Figure 3c C_M_ = 0.3, Figure 3d C_M_ = 0.2), the CVI was positive. But when the initial mannitol concentration C_M_ > C_ip_, this phenomenon did not appear. This phenomenon demonstrated that when cells absorb water and maintain their turgor pressure, the cells would exert a force on the chip.

### 4.4. Osmotic Pressure Model of Plant Cells

It is crucial for plants that osmolality, hydrostatic pressure, or turgor, and membrane tension are maintained in a fine balance [32]. Turgor pressure within a plant cell represents the key to mechanistically describe plant growth [33]. Rates of cell expansion must be matched by solute transport and its accumulation as the cell volume increases. Cell volume affects molecular crowding within the cytosol and intracellular ionic strength [34]. The central vacuole is the major compartment of osmotic water flow during plasmolysis but obviously, the abrupt change in protoplast size and shape impacts the subcellular architecture as a whole [35].

Aside from the central vacuole, a rigid cell wall is required for plasmolysis. This structure forms a solid shell encasing the osmo-sensitive and membrane-covered protoplast [36]. Cytoskeletal elements have been analyzed during a plasmolytic cycle in various plant species and cell types [37,38]. When the Hechtian strands have been broken (irreversibly damaged), the plasmolysis cannot be recovered. It is known that the two cytoskeletal elements are different differed in their re-organization patterns, and actin filaments and microtubules showed different responses to osmotic changes [37,39].

According to the changes of a BY-2 cell’s morphologies under different osmotic stresses (Figure 5 and Figure 6), we describe a structural model of a BY-2 cell under different osmotic pressures (Figure 7). This model is similar to the structure of a tire. We treat the structure inside of the cell membrane as a whole, including the vacuole. We consider the cell wall as the cover tire which shape is invariant, and the membrane as the inner tire. When C_M_ = C_ip_, the BY-2 cell is in isotonic state and the turgor is zero, the cell membrane has no force on the cell wall, just like a flat tire (Figure 7a), the inner tire does not exert any stress on the cover tire. There is free movement of water. There is no force on the surface of the chip by the cell wall. When C_M_ < C_ip_, intracellular osmolarity is lower than extracellular osmolarity, so BY-2 cell absorbs water, just like air is pumped into the inner tire, the inner tire pressurizes the cover tire, i.e., the cell membrane pressurizes the cell wall, and in turn the cell wall pressurizes the QCM chip (Figure 7b), sometimes there is a certain degree of shape change in the cell wall. When C_M_ > C_ip_, the intracellular osmotic pressure is higher than the extracellular osmotic pressure, so BY-2 cells lose water, the vacuoles and the cell membrane shrink, plasmolysis occurs, just like when pumping air out of the inner tire, the inner tire separates from the cover tire (Figure 7c). A big difference between the cell and the tire model is that the cytoskeleton in the cell is dynamic, while there is no skeleton in the tire, only air. Protoplast is not a single component structure, the structure of a plant cell is more complex, and the mechanical properties of plant cells are also very complex when osmotic pressure changes.

In the plasmolysis process, the protoplast shrinks away from the wall which is limited in many cell types by a physical linkage between the plasma membrane and the wall, a linkage that is strong enough to remain intact even as the bulk of the protoplast shrinks. The wall-plasma membrane anchoring sites are called Hechtian strands. Some, but not all, Hechtian strands form at plasmodesmata, where the plasma membrane is not only attached to the wall but transvacuolar, and is associated with the endoplasmic reticulum desmotubule within the plasmodesmata [40].

In isotonic solution, most of the BY-2 cells are in the initial state of plasmolysis, at this time, the cell membrane exerts no pressure on the cell wall, so the force on the chip is mainly due to the cell wall adhesion (Figure 7a). When the concentration of the solution is changed, the state of the cell is different. The effect of cell membrane on the cell wall and the force exerted on the chip is different. When C_M_ > C_ip_, high extracellular osmotic pressure causes vacuolar dehydration resulting in a decrease in the volume of the protoplast, the cell membrane still exerts no pressure on the cell wall, only the traction force of Hechtian’s strands would exert on cell wall caused by protoplast shrinkage (Figure 7b). With the increase of water loss inside the cell, the ion concentration in the cell would increase gradually, and the osmotic pressure inside and outside the cell would reach an equilibrium [41], while when C_M_ < C_ip_, the osmotic pressure would induce an increase in the internal volume of the protoplast, as the cell wall is only slightly deformed to maintain its shape to prevent this process. In that case, the protoplast has to exert force on the cell wall (Figure 7c) and thereby the QCM shows some different signals. The mechanisms through which plant cells control growth and shape are the result of the coordinated action of many events, notably cell wall stress relaxation and turgor-driven expansion [42]. Surrounded by a stiff wall, plant cells experience substantial turgor pressure, and growth occurs from the balance between this pressure and the loosening of the cell wall. Recent studies have shown that in wheat root hair under osmotic stress, exocytosis of wall material and its deposition, as well as callose synthesis, still occurred [43]. A comparison of the transcriptional changes induced by water stress resulted in the identification of genes whose expression was specifically affected in most species [44,45]. Some special chemicals should change the membrane permeability, in order to change the concentration of ions that induced initial plasmolysis [46]. So osmotic stress can affect the plant cell to adapt to its environment, through affecting the composition the cell wall.

## 5. Conclusions

The purpose of this work is to demonstrate the feasibility of using wide frequency QCM to measure and distinguish the different depths of cellular structures based on their viscoelastic differences. Plant cells are not homogeneous, and the outmost cell wall is normally most stiff and relatively invariant to maintain cells’ shape; the wall attached inside plasma membrane is a core structure directly subjected to the stress(es) from turgor pressure and actin cytoskeleton-generated forces; and the cytoskeleton is under constantly remodeling and reflecting the dynamic polymerization and depolymerization of cytoskeletal components of actin and microtubule. In this work, positively charged PDADMAC was used as an adhesion material to non-specifically adhere the plant cells and semiquantitative viscoelastic parameter of CVI_n_ at different overtone frequency (n) was used to characterize the adhesion of tobacco BY-2 cells and the subsequent responses under osmotic treatments of different mannitol concentrations. Our results indicated that lower overtone CVI_3_ can measure multiple structures of cell wall-plasma membrane-cytoskeleton allowing to detect the plasmolysis phenomenon; higher overtone CVI_9_ mainly detects the outmost cell wall and CVI_9_ monotonically increased with the decrease of mannitol concentration C_M_, i.e., the lower the C_M_, the stiffer the cell wall; CVI_7_ was always larger than CVI_9_ reflecting the plasma membrane stiffening under the action(s) of either turgor pressure and/or actin cytoskeleton-generated forces. In the future, we plan to modify the QCM chips with more specific molecules or materials which can interact with extracellular domains of transmembrane molecules such as formins, wall-associated kinases, cellulose synthase complexes, receptor-like kinases, and arabinogalactan proteins in conjugation with more powerful technique to simultaneously measure cells’ generated forces and viscoelastic parameters of elastic and loss moduli using both wall containing and wall free cells (protoplasts); then we would be able to compare the dynamic mechanical changes during adhesions of plant cells under the involvements of different transmembrane molecules. The final goal is to identify integrin-like force-sensing molecules of plant cells to study mechano-transduction of plant cells and their responses under various stresses.

## Figures and Tables

**Figure 1 biosensors-11-00136-f001:**
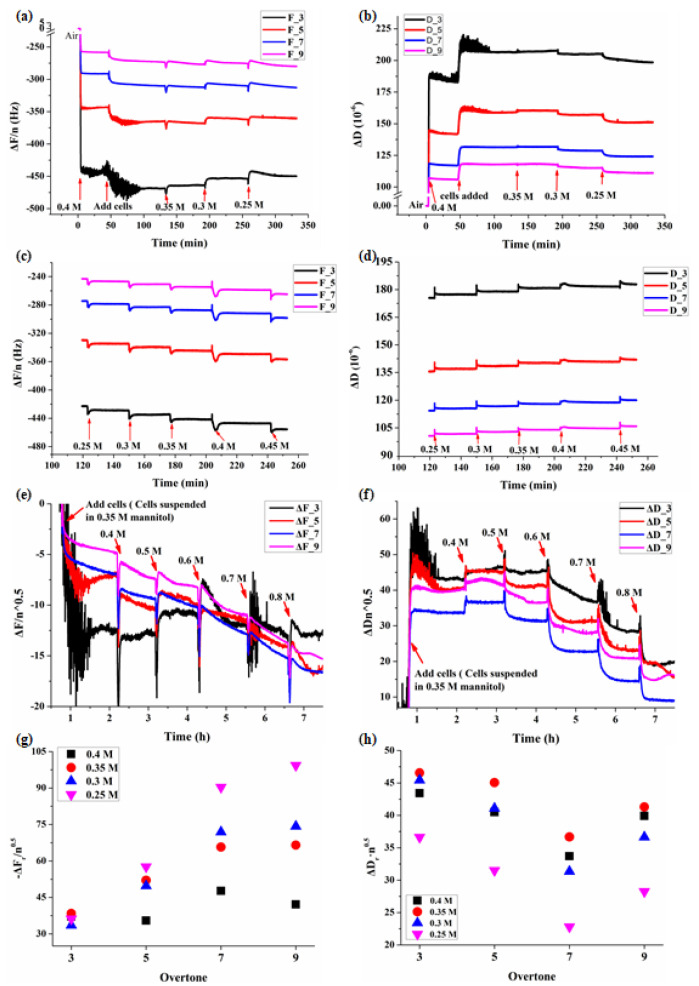
Typical QCM responses during the adhesions of BY-2 cells to (PDADMAC)/SiO_2_ crystal followed by osmotic treatments of different mannitol concentrations. (**a**,**b**) Changes in frequency and dissipation for each overtone during the adhesion of BY-2 cells in 0.4 M mannitol solution followed by hypotonic treatments of mannitol solutions. (**c**,**d**) The QCM-D responses of the reference (PDADMAC)/SiO_2_ crystal without cells in different concentrations of mannitol. (**e**,**f**) The QCM-D responses during the adhesion of BY-2 cells on (PDADMAC)/SiO_2_ crystal in 0.35 M mannitol followed by hypertonic treatments of mannitol solutions. (**g**,**h**) The relative changes ΔD_r_ = ΔD_BY-2_-ΔD_Mannitol_, ΔF_r_ = ΔF_BY-2_-ΔF_Mannitol_. ΔD_r_·n^0.5^ and -ΔF_r_/n^0.5^ were not constants for different overtones indicating the viscoelastic feature of living BY-2 cells.

**Figure 2 biosensors-11-00136-f002:**
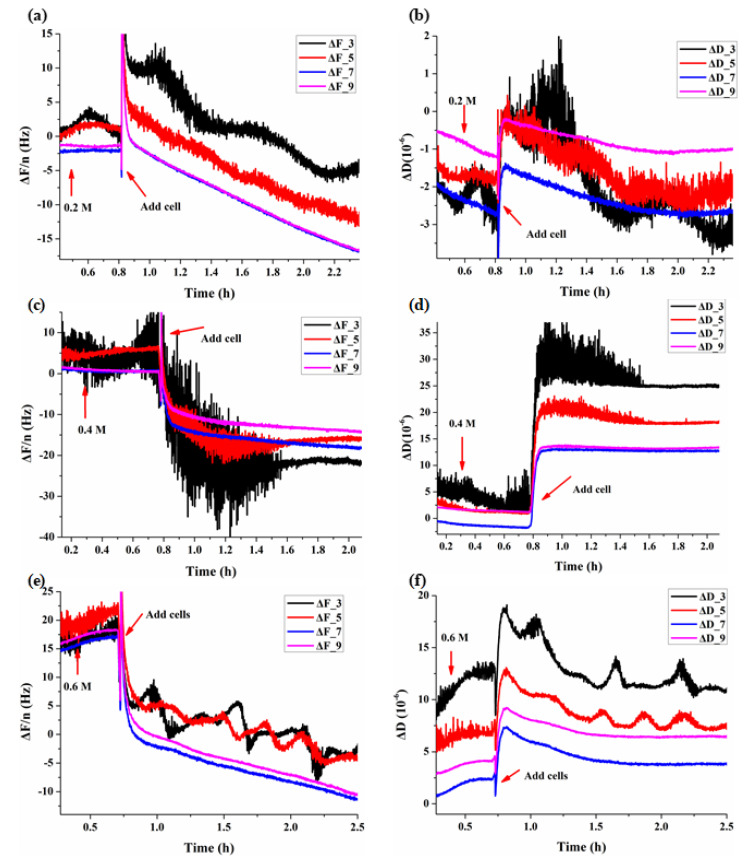
Dynamic QCM responses during the adhesions of BY-2 cells on (PDADMAC)/SiO_2_ crystals in different mannitol concentrations. (**a**,**b**) 0.2 M, (**c**,**d**) 0.4 M, (**e**,**f**) 0.6 M.

**Figure 3 biosensors-11-00136-f003:**
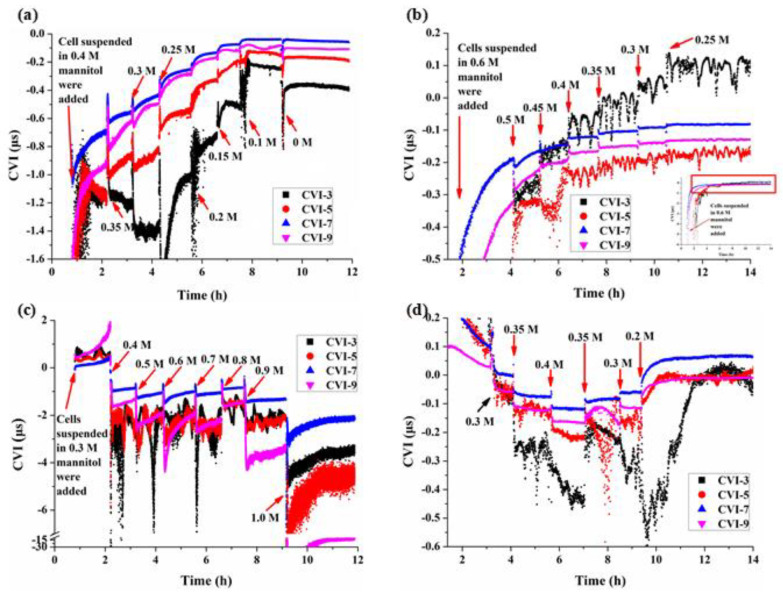
Dynamic viscoelastic changes of BY-2 cells under different osmotic conditions accompanied with changes in mannitol concentrations. (**a**) C_M_ dropped from 0.4 M to 0 M, (**b**) C_M_ dropped from 0.6 M to 0.25 M, (**c**) C_M_ increased from 0.3 M to 1.0 M, (**d**) C_M_ increased first from 0.2 M to 0.4 M then dropped to 0.2 M.

**Figure 4 biosensors-11-00136-f004:**
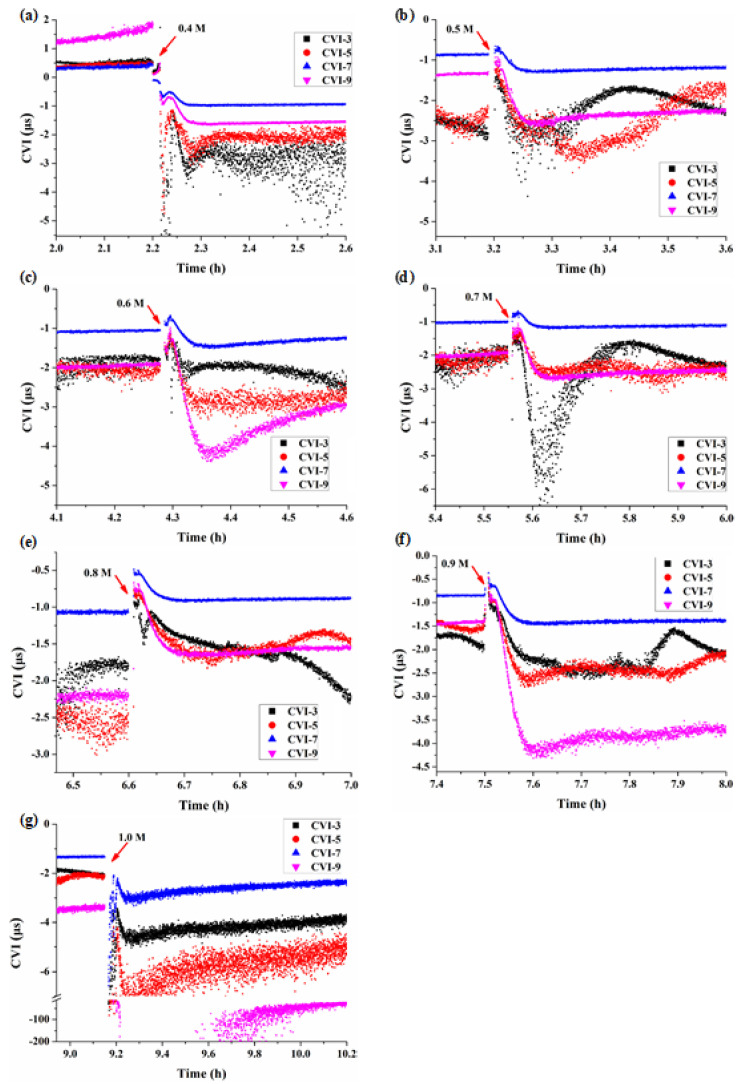
Expanded initial CVI responses of Figure 3c for BY-2 cells subjected to the hypertonic stresses of mannitol concentrations. (**a**) 0.4 M, (**b**) 0.5 M, (**c**) 0.6 M, (**d**) 0.7 M, (**e**) 0.8 M, (**f**) 0.9 M, (**g**) 1.0 M.

**Figure 5 biosensors-11-00136-f005:**
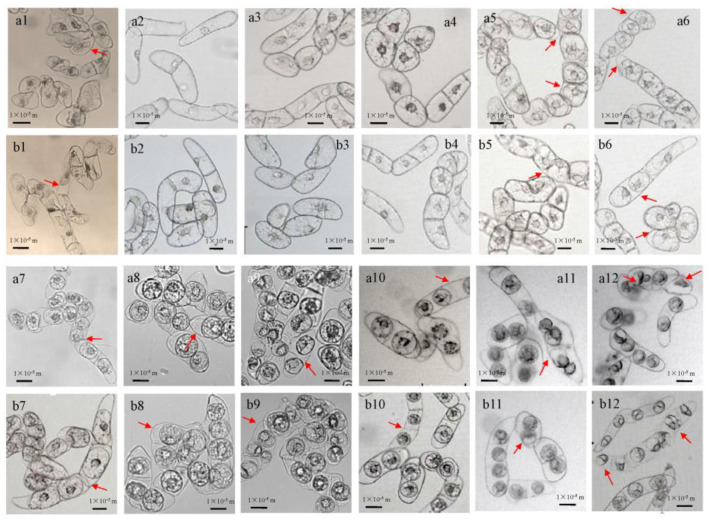
Morphological changes of BY-2 cells under different mannitol concentrations. 1: 0.0 M, 2: 0.1 M, 3: 0.2 M, 4: 0.3 M, 5: 0.35 M, 6: 0.4 M, 7: 0.5 M, 8: 0.6 M, 9: 0.7 M, 10: 0.8 M, 11: 0.9 M, 12: 1.0 M. (**a1**–**a12**): BY-2 cells were suspended in centrifuge tubes containing mannitol solution for 2 h first, and then immobilized on PDADMAC modified glass slides. (**b1**–**b12**): BY-2 cells were immobilized on PDADMAC modified glass slides and then stayed in the mannitol solution for 2 h. The large sacs containing of fluid shown are vacuoles.

**Figure 6 biosensors-11-00136-f006:**
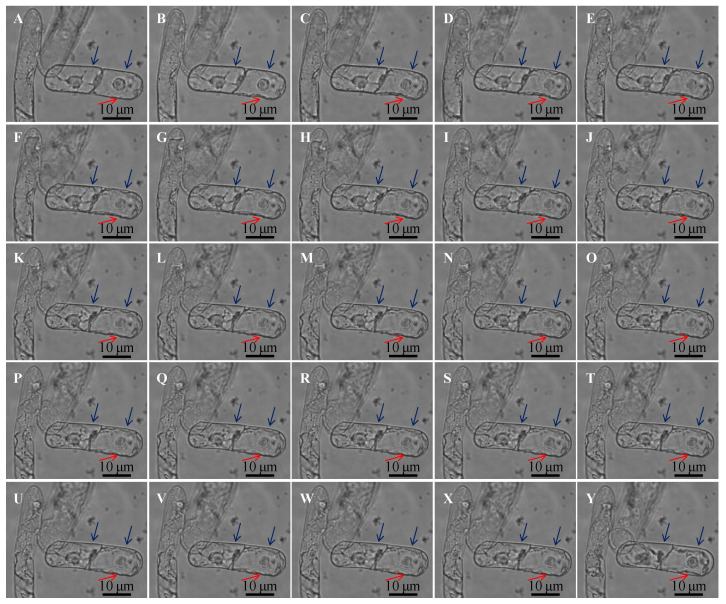
Morphological changes of BY-2 cells at the moment of osmotic pressure change. (**A**): Morphology of BY-2 cells in isosmotic solution before adding hypertonic solution. (**B**–**X**): Morphologies of BY-2 cells at intervals of 30 s after 1 M mannitol solution was added and the concentration of the solution changed to 0.4 M. (**Y**): Morphology of BY-2 cells at about 30 min after 1 M mannitol solution was added and the concentration of the solution changed to 0.4 M.

**Figure 7 biosensors-11-00136-f007:**
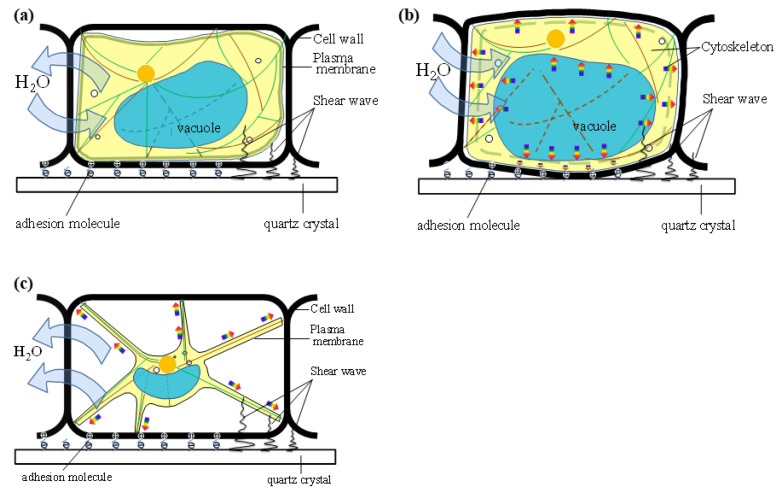
Schematic of the BY-2 cell morphologies under different osmotic stresses. (**a**) C_M_ = C_ip_, BY-2 cell in normal state and the turgor is zero, the cell membrane exerts no force on the cell wall. (**b**) C_M_ < C_ip_, BY-2 cell absorbs water and the cell membrane would pressurize the cell wall, and in turn the cell wall pressurizes the QCM chip, there is a certain degree of shape change in the cell wall. (**c**) C_M_ > C_ip_, BY-2 cell loses water. As the loss of water, the vacuoles and the cell membrane shrink and plasmolysis would occur.

**Table 1 biosensors-11-00136-t001:** Some of the Major Progresses of QCM Technology in Cellular and Biomolecular Assays.

Application Examples	Technical Feature	Technical Characteristics
High-throughput QCM (HQCM) chip for the study of cell-drug interactions	Configuration of interference-free, negligible installation-induced stress	Fabricated using independent yet same-batch quartz crystal resonators within a common glass substrate through the rigid (quartz)-soft (silicone)-rigid (glass) structure [17]
Cell−material interactions	Tuning the surface free energy of materials	The results demonstrated that tuning the surface free energy of materials is a useful strategy for selectively promoting eukaryotic cell adhesion and preventing bacterial adhesion [18]
Oxidative stress on the viscoelastic properties of pre-osteoblast cells	Cell recovery from oxidative stress indicated by viscoelasticity	While a return to baseline values of the energy dissipation (ΔD) response was obtained at 325 min (recovery point) after the incubation of cells with 25 μM H_2_O_2_, higher concentrations (50 μM–10 mM) exhibited no recovery [19]
Evaluation of cell migration	Real time monitoring of cell regulatory volume decrease (RVD)	RVD was tracked via analyzing frequency shifts during the cell swelling and shrinkage and the results showed that the level of RVD for MCF-7 cells and MDA-MB-231 cells was 32.8% and 49.7% indicating their difference in migration [9]
Attachment of suspension cells	Mussel-inspired polydopamine (PDA) as the coating to attach suspension cells	The PDA-based suspension-cell QCM biosensor showed high degree of repeatability and stability as well as low nonspecific binding to the irrelevant protein [20]
Conventional adhesion material for plant cells	Poly-L-lysine (PLL)-3-hydroxyphenyl-boronic acid (3-PBA) composite was prepared via cyclic voltammetry	Synergistic effect of 3-PBA’s interactions with glycosyl compounds on the cell wall and PLL’s electrostatic interactions with the negatively charged cell wall promoted the adhesion of plant cells [21]
Direct, label-free detection of biomarkers in a large amount of contaminants	Ultrahigh-frequency, wireless MEMS QCM biosensor	Nonspecific adsorption of contaminants in analyte solution was avoided at ultrahigh-frequency (∼576 MHz) [22]
Selecting highly dissipative probes for ultrasensitive DNA detection	Liposomes as probes for energy-dissipation enhancement	The parameter of dissipation capacity, the ability to dissipate acoustic energy at the level of a single molecule/particle was introduced [23]

## Data Availability

The authors confirm that the data supporting the findings of this study are available within the article.

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
