# Peer review of "Quartz Crystal Microbalance with Dissipation Monitoring of Dynamic Viscoelastic Changes of Tobacco BY-2 Cells under Different Osmotic Conditions"

_biosensors, 2021, doi:10.3390/bios11050136_

Round 1
Reviewer 1 Report
The manuscript entitled: "Quartz Crystal Microbalance with Dissipation Monitoring of 2 Dynamic Viscoelastic Changes of Tobacco BY-2 Cells under 3 Different Osmotic Conditions" represents an interesting approach, but needs some improvements, as follows:
- In the Introduction, there are repeated twice the characteristics of Quartz crystal microbalance (QCM) technology, instead of providing a table to review the use of this technique and its performances in the last years.
- There is no clear emphasizing of the importance of viscoelastic changes of different cellular structures and the health of the plants, or other characteristics.
- The abbreviations has to be written after the IUPAC name, no vice-versa.
- All figuresare too small, not appropriately presented, so that they cannot be understood. I suggest splitting them on different criteria (for instance Figure 1 g and h might be a different Figure).
- Figure 3 is also not appropriately presented.
- The discussion needs a better structure, so that the type of stress to be clearly in relationship with the final response.
- References, with a few exceptions are old, and this is not a plus for the paper.
Reviewer 2 Report
In this manuscript, the Quartz crystal microbalance was used for the dynamic monitoring of adhesions of living tobacco BY-2 cells onto positively charged PDADMAC modified QCM surfaces under different concentrations of mannitol and the subsequent effects of osmotic stresses. The cell viscoelastic index was used to characterize the viscoelastic properties of plant cells under different osmotic conditions. There are some problems in the article, and the details are as follows:
- Since plant cells are different from animal cells, and plant cells lack the growth property of autonomous adhesion to the substrates, so what is the significance of monitoring the adhesion of plant cells with Quartz crystal microbalance? The significance should be explained in the text.
- The dynamic changes of cell structure, such as the plasmolysis phenomenon, can be observed through the optical microscope, and the results are as clear and intuitive as the microscope results in this paper. Compared with the microscope detection, what is the advantage of the Quartz crystal microbalance in detecting changes of cell structure?
- Many things were not described clearly in the text, for example, the relationship between cell adhesion and cell mechanical properties, and how could the structure of cell wall and cytoskeleton be identified from the results of QCM?
Round 2
Reviewer 1 Report
The Authors improved their manuscript but I consider that the paper will be of better quality if a table will be introduced in the Introduction section to comparatively review the use of this technique and its performances in the last years.
Reviewer 2 Report
The authors have satisfactorily responded to all the questions and made the necessary changes to the manuscript. I have no further questions and suggest the acceptance of the revised manuscript.
